# Slow Freezing of Preserved Boar Sperm: Comparison of Conventional and Automated Techniques on Post-Thaw Functional Quality by a New Combination of Sperm Function Tests

**DOI:** 10.3390/ani13182826

**Published:** 2023-09-06

**Authors:** Felipe Pezo, Fabiola Zambrano, Pamela Uribe, André Furugen Cesar de Andrade, Raúl Sánchez

**Affiliations:** 1Escuela de Medicina Veterinaria, Facultad de Recursos Naturales y Medicina Veterinaria, Universidad Santo Tomas, Santiago 8370003, Chile; felipezo1982@gmail.com; 2Laboratory of Reproductive Physiopathology, Center for Translational Medicine (CEMT-BIOREN), Temuco 4811230, Chile; fabiola.zambrano@ufrontera.cl (F.Z.); pamela.uribe@ufrontera.cl (P.U.); 3Department of Preclinical Sciences, Faculty of Medicine, Universidad de La Frontera, Temuco 4811230, Chile; 4Department of Internal Medicine, Faculty of Medicine, Universidad de La Frontera, Temuco 4811230, Chile; 5Department of Animal Reproduction, School of Veterinary Medicine and Animal Science (FMVZ), University of São Paulo (USP), Pirassununga 13635-900, SP, Brazil; andrefc@usp.br

**Keywords:** boar sperm, cryopreservation, slow freezing, conventional freezing, automated freezing

## Abstract

**Simple Summary:**

The slow freezing of boar spermatozoa is carried out by exposure to liquid nitrogen vapors (conventional) or by using automated equipment that controls the temperature drop curve. In both cases, freezing/thawing causes a reduction in sperm functionality. Therefore, the objective of this study was to compare both freezing techniques through a panel of evaluations of sperm function by CASA and flow cytometry technologies. We observed the negative effect of post-thawing incubation time on sperm motility. In addition, there was a significant decrease in the levels of free radicals, reduced lipid peroxidation, and a loss of cholesterol and markers of apoptosis, which have an impact on the better conservation of the motility of frozen boar spermatozoa with automated equipment. For this reason, the control of the temperature drop curve allows sperm functionality to be preserved. No significant differences were observed for some parameters associated with the state and preservation of the membranes.

**Abstract:**

The slow freezing of boar sperm is the only way to preserve genetic material for extended periods; this can be achieved with exposure to liquid nitrogen vapors (conventional) or by using automated freezing equipment. The aim was to compare the effect of both techniques on post-thaw functionality. Boar sperm devoid of seminal plasma and resuspended in lactose-egg yolk-glycerol medium were cryopreserved. Conventional: straws were exposed to LN_2_ vapors; automated: using a drop curve of −39.82 °C·min^−1^ for 113 s from −5 to −80 °C during the critical period; and subsequent immersion in NL_2_. Cell viability, cholesterol flow, mitochondrial membrane potential (MMP), lipid peroxidation, peroxynitrite, superoxide anion levels, phosphatidylserine translocation, and caspase activation were evaluated by flow cytometry. In addition, total motility (TM) and progressive motility (PM) were determined by the SCA system immediately (T0), 60 (T60), and 120 min (T120) post-thawing. Automated freezing significantly reduces cholesterol flow and free radical and lipid peroxidation levels, making it possible to preserve motility for 120 min of incubation. At the same time, viability, acrosome integrity, MMP, and caspase activation did not differ from the conventional technique. In conclusion, controlling the temperature drop curve using automated freezing equipment reduces oxidative/nitrosative stress, preserving membrane fluidity and sperm motility.

## 1. Introduction

In pigs, artificial insemination (AI) is mainly performed with semen stored at 17 °C in a suitable dilution medium. However, the main limitation of this reproductive biotechnology, associated with the preservation of gametes, is the short period for which it can be stored (5–7 days) [1]. The cryopreservation of the male gamete has several advantages because the metabolic rate decreases with temperature; therefore, thermally driven chemical reactions do not occur in biological systems at cryogenic temperatures [2]. The freezing and thawing processes have a negative impact on the quality of the sperm sample; these cryoinjuries affect the integrity of the sperm plasma membrane, acrosome, and nucleus, as well as mitochondrial function and sperm motility [3], which could eventually affect in vivo results, which are usually evaluated based on pregnancy rates, return rates, and litter size, among other reproductive variables.

Currently, two principal techniques are applied to the cryopreservation of domestic animal sperm: slow freezing and vitrification. Vitrification consists of plunging the sperm samples directly into a suitable extender into liquid nitrogen (LN_2_), and is a fast, simple, and cost-effective technique that is well developed to cryopreserve human sperm [4]; however, in boar, motile sperm were not observed after vitrification, allowing its use only in intracytoplasmic sperm injection (ICSI) [5,6]. Slow freezing is widely used to cryopreserve boar sperm and can be performed by controlling or not controlling the temperature descent curve. The conventional slow freezing technique consists of placing semen straws (10–20 min) on racks suspended a few centimeters (3–5 cm) above LN_2_, which are kept in a Styrofoam box and frozen in liquid nitrogen vapors [7,8], the approximate temperature range of which is between −60 and −70 °C/min [9]. The automated slow freezing technique, using a biofreezer, controls the freezing rate, which is extremely important in maintaining a wide variety of cellular functions. In fact, according to Mazur [10], the optimal freezing rate should be slow enough to allow the sufficient dehydration of the sperm cell and prevent intracellular ice crystals from forming, and fast enough to avoid the sperm being exposed to hyper-saturated solutions as ice crystals form in the extracellular environment.

In boar sperm, it has been determined that freezing rates of either 20, 40, or 60 °C/min, which are achieved using automated freezing, resulted in the improvement of post-thaw motility (45 versus 52% for a conventional and automated freezing rate of 40 °C/min, respectively), viability, the acrosomal integrity of live sperm, and the mitochondrial membrane potential of live sperm compared to conventional slow freezing [11]. Different groups of researchers around the world use automated freezing systems and a temperature drop rate of −40 °C/min between −5 and −80 °C, which is when the freezing of the different cellular components occurs [12,13,14]. However, these systems are expensive, which means that many researchers and clinicians continue to perform the conventional freezing technique in a Styrofoam box, where there is no control of freezing rates, and there is also no standardization regarding the height where the straws are positioned and the time for which they are exposed to liquid nitrogen vapors [15,16,17,18].

Few comparative studies have been developed to determine if the control of the temperature drop curve improves the in vitro or in vivo results of sperm function in domestic animals. The aim of this study was to compare the two most used techniques in the slow freezing of boar sperm and the effect on different sperm parameters associated with the state of the sperm membranes, oxidative stress, and the presence of apoptosis markers.

## 2. Materials and Methods

### 2.1. Ethical Declarations

All protocols were approved by the Scientific Ethics Committee of the Universidad de La Frontera, Chile, and carried out according to Chilean Law No. 20.380 for the Protection of Animals.

### 2.2. Study Sites

Cryopreservation protocols and motility assessments were conducted at the Sperm Biology and Conservation Laboratory in the Center for Translational Medicine (CEMT-BIOREN). In addition, flow cytometry tests were performed at the Scientific and Technological Bioresource Nucleus (BIOREN). All these facilities are part of the Universidad de La Frontera (Chile).

### 2.3. Reagents

Unless otherwise indicated, all reagents were purchased from Sigma (St. Louis, MO, USA). All the solutions were prepared using water from a Milli-Q Synthesis system (Millipore, Bedford, MA, USA).

### 2.4. Semen Samples

Twenty ejaculates from eight different boars with proven field fertility were collected. The boars (Landrace) were housed at Agricultural and Livestock Society Pehuén Ltd.a. in Victoria, Chile. The sample was collected without the gel portion of the semen using the glove-hand technique, in an isolated receptacle covered with gauze. AndroStar Plus^®^ (Minitüb, Tiefenbach, Germany) was used as a dilution and refrigeration medium and only the sperm-rich fraction of each ejaculate was diluted and split into commercial semen doses of 80 mL, with 30 × 10^6^ sperm /mL. Semen doses were kept at 17 °C and transported to the laboratory within one hour. To reduce variations between experiments, all semen samples were kept in a refrigerated chamber at 17 °C for 24 h [19], and only samples with a total sperm motility of over 80% and viability over 85% were used [20].

### 2.5. Sperm Cryopreservation Protocol and Experimental Design

Boar sperm were cryopreserved as described in Estrada et al. [20]. Lactose egg yolk (LEY; (80% (*v*/*v*) β-lactose 250 mM, 20% (*v*/*v*) of chicken egg yolk, and 100 mg/L of kanamycin sulfate) and LEY plus 2% glycerol and 0.5% Equex (LEYGO) were used as freezing media. The semen doses diluted in the AndroStar Plus^®^ medium were centrifuged at 600× *g* for 5 min at 17 °C, and the sperm pellets were resuspended in LEY medium (1.5 × 10^9^ sperm/mL) and cooled down from 17 to 5 °C in one hour. After that, sperm suspensions were diluted with LEYGO medium (1 × 10^9^ sperm/mL). The strawing was carried out at 5 °C using 0.25 mL straws. Two different freezing protocols were used: Conventional, exposing the straws placed upon racks suspended at 4 cm for 15 min to LN_2_ vapors on a Styrofoam box. Automated, where freezing is achieved with automated equipment (Icecube11 XS) using a temperature drop curve of 6 °C·min^−1^ for 100 s (from 5 to −5 °C); −39.82 °C·min^−1^ for 113 s (from −5 to −80 °C); holding time at −80 °C for 30 s; and −60 °C·min^−1^ for 70 s (from −80 to −150 °C). The straws were stored in LN_2_ for further analysis. Beltsville Thawing Solution (BTS) at a ratio of 1:4 was used as a thawing medium; the restoration of sperm metabolism was completed in a thermo-regulated water bath at 38 °C for 20 s.

### 2.6. Analysis of Sperm Motility

Sperm motility was tested with a sperm class analyzer (SCA) system. After thawing, sperm suspensions were kept in a chamber at 38 °C, and the analysis was performed immediately (T0), 60 (T60), and 120 (T120) min post-thawing. Then, 6 μL of each sample was placed on a 20 µm depth counting chamber heated at 38 °C [21]. The SCA system is based on the analysis of 25 consecutive digitized photographic images obtained from a single field under negative phase contrast and at 10× magnification. The parameters were adjusted to boar sperm, and the particle area was set at 10–80 μm [22]. Sperm with an average path rate (VAP) equal to or higher than 10 μm/s were classified as immotile. In addition, sperm with >45% straightness (STR) and VAP ≥ 45 μm/s were considered progressively motile. Five fields were taken for each experimental group, analyzing a minimum of 500 sperm per replicate.

### 2.7. Flow Cytometry Analyses

Fluorescence analysis was performed in a FACSAria fusion flow cytometer (Becton, Dickinson and Company, BD Biosciences, San Jose, CA, USA) equipped with 405, 488, and 640 nm lasers. Samples were acquired and analyzed using the software FACSDiva™ v. 8.0.2 (Becton, Dickinson, and Company). Data from 10,000 sperm events were recorded. The sperm population was selected by adding Hoechst 33342 (ThermoFisher, Waltham, MA, USA), a cell-permeant nuclear stain that bound to double-stranded DNA.

#### 2.7.1. Mitochondrial Membrane Potential, Peroxynitrite Levels, and Cell Viability

A multiparametric analysis was performed according to Pezo et al. [23], with slight modifications, to evaluate cell viability (propidium iodide, PI; final concentration: 12 µM), peroxynitrite levels (Dihydrorhodamine 123, DHR; final concentration: 1 µM), and mitochondrial membrane potential (Tetramethylrhodamine, TMRM; final concentration: 250 µM). Briefly, DHR, TMRM, PI, and Hoechst 33342 (final concentration: 1.69 μM) were added to 1 mL of BTS containing 2 × 10^6^ frozen-thawed sperm. Sperm were incubated at 38 °C for 20 min and then centrifuged at 800× *g* for 5 min. Finally, pellets were resuspended in 500 µL of BTS. The percentages of viable sperm (PI^−^) and mean fluorescence intensities (MFI) of DHR, and TMRM in the viable sperm population, were determined by flow cytometry.

#### 2.7.2. Superoxide Anion Levels and Acrosome Membrane Integrity

Dihydroethidium (DHE; final concentration: 40 µM) and *Lectin* from *Arachis hypogaea* (PNA-FITC; final concentration: 0.3 µg/mL) were used to determine ·O_2_- levels and acrosome integrity. For 20 min, 400 µL of BTS containing 2 × 10^6^ frozen-thawed sperm were incubated with DHE, PNA-FITC, PI, and Hoechst 33.342 at 38 °C. After that time, the sperm suspension was centrifuged at 800× *g* for 5 min to then discard the supernatant and resuspend the sperm pellet in 500 µL of BTS. For statistical analysis, we considered the MFI of DHE in the viable sperm population (PI^−^) and the percentages of viable sperm with an intact acrosome (PNA^−^/PI^−^).

#### 2.7.3. Membrane Lipid Peroxidation

C11-BODIPY581/591 (BODIPY; final concentration: 5 µM) was used to determine the lipid peroxidation of plasma membranes. For 30 min, 400 µL of BTS containing 2 × 10^6^ frozen-thawed sperm were incubated with BODIPY, PI, and Hoechst 33,342 at 38 °C. After that time, the sperm suspension was centrifuged at 800× *g* for 5 min to then discard the supernatant and resuspend the sperm pellet in 500 µL of BTS. For statistical analysis, we considered the MFI of BODIPY in the viable sperm population (PI^−^).

#### 2.7.4. Modifications of Plasma Membrane Lipids

Merocyanine-540 (M-540; final concentration: 250 µM) and SYTOX Green (final concentration: 0.04 µM) were used to determine the modifications of plasma membrane lipids in the living population. For 20 min, 500 µL of BTS containing 2 × 10^6^ frozen-thawed sperm were incubated with M-540, SYTOX and Hoechst 33,342 at 38 °C. After that time, the sperm suspension was centrifuged at 800× *g* for 5 min to then discard the supernatant and resuspend the sperm pellet in 500 µL of BTS. For statistical analysis, we considered the percentages of viable sperm without membrane lipid disorder (M-540-).

#### 2.7.5. Phosphatidylserine Translocation

The translocation of phosphatidylserine (PS) residues to the outer leaflet of the plasma membrane were detected with an Annexin Apoptosis Detection Kit (Sigma, Madrid, Spain). The protocol was performed according to the manufacturer’s instructions, with slight modifications. For this assay, 1 μL of Annexin V, PI, and Hoechst 33,342 were added to 450 μL of binding buffer containing 2 × 10^6^ frozen-thawed sperm. After 15 min of incubation in the dark at room temperature, the sample was centrifuged at 800× *g* for 5 min. Pellets were resuspended in 500 µL of BTS, and the percentages of viable sperm with PS translocated (Annexin^+^/PI^−^) were determined by flow cytometry.

#### 2.7.6. Caspase 3/7 Activation

Caspase 3/7 activity in frozen-thawed sperm was evaluated using a FAM-FLICA^®^Caspase Assay (Immunochemistry Technologies, LLC, Davis, CA, USA), according to the manufacturer’s instructions. Briefly, 5 × 10^5^ of frozen-thawed sperm was placed in 300 µL of 30× FLICA solution. Incubation was at 37 °C for 1 h and protected from light. PI and Hoechst 33,342 were added 15 min before the end of the incubation with the FLICA solution. Subsequently, the sample was centrifuged at 800× *g* for 5 min. Finally, pellets were resuspended in 500 µL of BTS, and the percentages of viable sperm with an increase in caspase activation (FLICA^+^/PI^−^) were determined by flow cytometry.

### 2.8. Statistical Analyses

GraphPad Prism^®^ v. 5.0 (GraphPad Software, San Diego, CA, USA) was used for statistical analysis. D’Agostino’s K2 test was applied to evaluate the Gaussian distribution. A two-way ANOVA of repeated measures and Bonferroni’s post-test were used to analyze the effect of incubation time on sperm motility (*n* = 20). In addition, a *t*-test was applied at each of the times evaluated for motility parameters, and the data from flow cytometry (*n* = 12) experiments were analyzed using a *t*-test, comparing the two freezing techniques. Values of *p* < 0.05 were considered statistically significant, and the results are expressed as mean ± standard deviation (SD).

## 3. Results

### 3.1. Motility

The two-way ANOVA revealed the negative effect of sperm incubation time at 38 °C throughout the 120 min on total (TM) and progressive motility (PM), independent of the freezing technique. When both techniques were compared in each of the evaluated times (*t*-test), for TM (Figure 1A), the use of the automated system was significantly superior to the conventional technique immediately after thawing (T0; *p* = 0.0114) and at 120 (T120; *p* = 0.0192) min of incubation at 38 °C. For PM (Figure 1B), using the automated system significantly preserved this parameter throughout the 120 min of incubation at 38 °C (T0, *p* = 0.0077; T60, *p* = 0.0380; T120, *p* = 0.0440).

### 3.2. Flow Cytometry Analysis

A comparison of the two freezing techniques revealed a significant reduction in superoxide anion (Figure 2B) and peroxynitrite levels (Figure 2C) with the automated technique. This effect on free radical levels positively impacted the reduction in lipid peroxidation (Figure 2A) and with it, capacitation-type changes, since there was a significant reduction in cholesterol flow (Table 1). However, despite the decrease in oxidative/nitrosative stress as a result of controlling the temperature decrease curve, the latter had no impact on the better maintenance of sperm membranes, since no significant differences were detected between the two techniques for plasma and acrosomal membrane integrity (Table 1) and mitochondrial potential (Figure 2D). Additionally, while employing automated freezing, a substantial decrease in phosphatidylserine translocation was noted when the two procedures were compared for the presence of apoptotic indicators (Table 1). Caspase activity did not present significant differences between both groups.

## 4. Discussion

Male gamete cryopreservation continues to be a challenge for different groups of researchers seeking to improve this reproductive biotechnology. Nowadays, boar sperm can be frozen with exposure to NL_2_ vapors (conventional) or by using an automated system to control the temperature drop curve. In our comparative study, the automated group shows a reduction in free radicals’ levels and with it, a significant decrease in lipoperoxidation, which had a positive impact on reducing cholesterol flow, decreasing the translocation of phosphatidylserine (PS), and preserving better motility over 120 min (T120) of incubation at 38 °C. However, the automated freezing technique did not significantly reduce caspase activation and did not positively impact the sperm membrane states compared to the conventional technique. It is well known that one of the variables to take into consideration during the freezing of any cell line is the temperature drop curve due to the heterogeneity of the sperm membrane that implies different ice nucleation points [24]. In addition, it is known that boar sperm are more sensitive to heat shock due to the low cholesterol and high protein content [25,26].

In boar sperm, when conventional slow freezing is used, with exposure to liquid nitrogen vapors, the freezing rate is approximately −60 °C/min [27], which is conditioned by the height and time that the straws are exposed to liquid nitrogen vapors. Conversely, automated slow freezing usually employs a curve using a drop rate of −40 °C/min between −5 and −80 °C [28,29], a critical period in which the different cellular components freeze [30]. The rate at which ice forms during sperm cryopreservation is strongly dependent on the freezing protocol; slow freezing relies on relatively slow cooling rates (0.1–50 °C/min) and low concentrations of cryoprotective agents; such conditions allow cells to lose intracellular water during freezing in a controlled manner, avoiding intracellular ice formation [31]. Sperm biomolecules may undergo conformational and membrane phase changes by lowering the temperature, inducing cell dehydration. This re-organization of membrane domains during cooling and freezing may cause malfunctioning after thawing, a detrimental effect that we observed in motility patterns for 120 min of incubation at 38 °C.

According to the state of the art, there is only one comparative study on boar sperm, which concluded that the controlled freezing rate (freezing rates of 20, 40, or 60 °C/min between −6 and −140 °C) resulted in the substantial improvement of post-thaw sperm motility, live sperm, live intact acrosome, and live sperm with high MMP compared to the conventional technique [11]. By contrast, in our study, for viability, even when a trend is observed, it is not significant, and no positive effect on acrosome intactness or MMP was observed compared to the conventional technique. Studies on different species of domestic animals have tried to determine if automated freezing is superior to conventional freezing, and the results presented are controversial. In stallions, it was determined that only progressive motility improved significantly when using the automated freezing technique, applying a drop temperature of −10 °C/min until −110 °C/min [32]. This is consistent with what was reported for the same species comparing the two techniques, with the automated system being better for motility, viability, and plasma membrane integrity after 24 h of cooling at 16 °C [33]. However, it was later suggested that the two techniques did not differ in the sperm viability and conception rates, indicating that the two methodologies can be safely used in AI programs. The automated system was set at −10 °C/min between 5 °C and −60 °C, at a rate of −8 °C/min between −60 °C and −100 °C, 5 °C and −60 °C, and at a rate of −8 °C/min between −60 °C and −100 °C [8]. In bull sperm, the TK 4000^®^ system using a freezing rate of −15 °C/min versus the conventional technique (Styrofoam box) allows for the better preservation of total and progressive motility and different kinetic parameters. Still, there is no effect on membrane and acrosomal integrity, MMP, or H_2_O_2_ levels [34].

In general, freezing the male gamete includes a series of steps that enable the cell to adapt to temperature changes. It has been identified that the cooling rate is a highly influential factor affecting post-thaw sperm quality [35]. The cooling rate after ice nucleation is also considered an important factor, and a cooling program with low cooling rates directly after nucleation and higher cooling rates at lower subzero temperatures was found to yield optimal results [36]. Sperm experience stress in the form of cold, osmotic, and mainly oxidative stress during cryopreservation, which reduces sperm viability and fertility. This is why the use of a biofreezer would allow us to obtain better results, since it would allow us to control the rate of temperature decrease, particularly in the period in which the cellular components reach their nucleation point.

Conventionally cryopreserving boar sperm causes an increase in the levels of free radicals that induce cell aging, and with it, a series of deleterious effects that alter cell function [18]. In automated freezing, there is also evidence of an aging process following prolonged cryostorage in LN_2_ due to a significant increase in the sperm cryo-susceptibility to induced lipoperoxidation and DNA fragmentation [37]; this could explain why cryostorage for longer than two years can be particularly detrimental to post-thaw motility [38]. Our results show that using automated freezing equipment will reduce oxidative/nitrosative stress and thereby the externalization of PS, as well as significantly preserving sperm motility immediately after thawing or throughout 120 min of incubation, respectively. PS oxidation is a sufficient component of the apoptotic machinery, ultimately contributing to PS externalization [39]; this exposure on the outer leaflet of the plasma membrane has several potential biological consequences, and one of them could be leukocyte activation.

Despite the better conservation of motility observed using the automated technique, in both groups it was evidenced that incubation time negatively affects motility for 120 min at 38 °C. This is consistent with different studies on swine species, where a recent study showed that boar sperm viability and motility considerably decrease after 300 min of incubation at 38 ℃ [40]. For this reason, a filtration technique (Sephadex) is recommended, which removes dead or damaged sperm after thawing [41].

The implementation of an automated freezing system requires a high initial cost and maintenance of the equipment; therefore, conventional freezing is the most used freezing technique under field conditions. For research groups, the challenge with slow freezing cryopreservation approaches is to find the cell-specific optimal cooling rate resulting in maximal cryo-survival, given that the biophysical properties that increase post-thaw survival are poorly understood. All those strategies targeting cryopreserved boar sperm must result in acceptable conception rates and produce 11 to 12 piglets/litter to compete with conventionally cooled semen.

## 5. Conclusions

Nowadays, conventional and automated freezing techniques are used for boar sperm cryopreservation. Regardless of the technique, exposure to subzero temperatures reduces cell function, which increases with incubation time; however, automated freezing reduces free radical levels and lipid peroxidation and phosphatidylserine translocation, preserving motility over time. We observed the sensitivity of a cell to temperature changes, the control of which positively affects cell function. Nevertheless, despite automated freezing being superior in some of the parameters evaluated, we believe that conventional freezing is a cost-effective alternative. However, further studies are needed to enable its standardization, since there is a high variability in height and time in which the straws are exposed to liquid nitrogen vapors.

## Figures and Tables

**Figure 1 animals-13-02826-f001:**
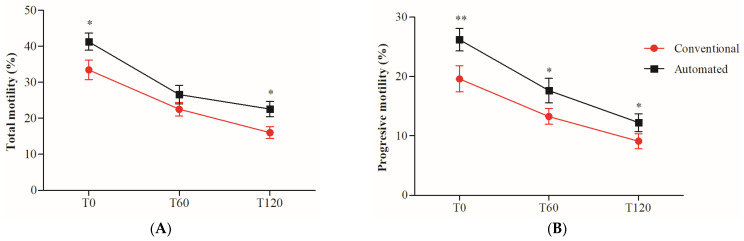
Percentage of total (TM) and progressive (PM) motility in cryopreserved boar sperm using conventional or automated freezing techniques. TM (**A**) and PM (**B**) were evaluated at 0 (T0), 60 (T60), and 120 (T120) minutes of incubation at 38 °C. The results are presented as mean ± SD (*n* = 20). * Indicates significant differences between the two freezing techniques (* *p* < 0.05; ** *p* < 0.01).

**Figure 2 animals-13-02826-f002:**
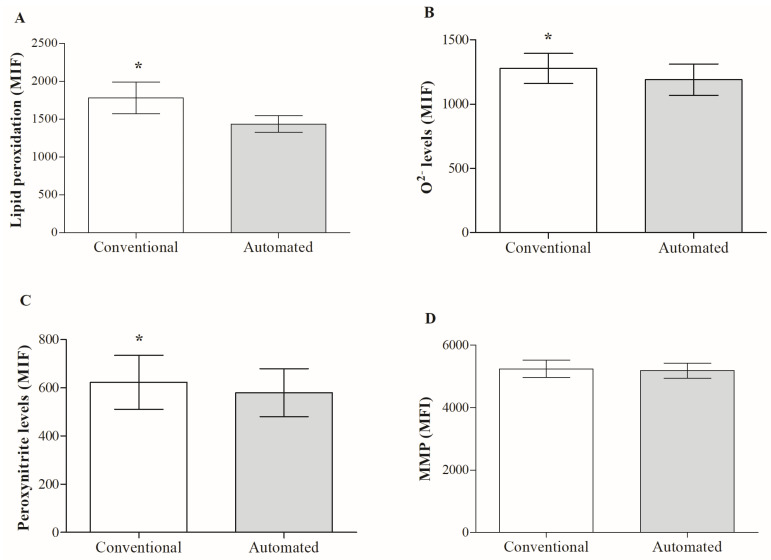
Effect of the freezing technique on oxidative/nitrosative stress markers and mitochondrial membrane potential (MMP) of frozen/thawed boar sperm. (**A**) lipid peroxidation, (**B**) ·O^2−^ and (**C**) peroxynitrite levels, and (**D**) MMP. Results are expressed as mean ± SD (*n* = 12). Abbreviations: MFI: mean fluorescence intensity; (*) indicates significant differences between the two freezing techniques (*p* < 0.05).

**Table 1 animals-13-02826-t001:** Effect of boar sperm freezing technique sperm on the states of membranes and the presence of apoptosis markers.

Freezing Technique	% Membrane Integrity (PI^−^)	% Acrosome Integrity (PNA^−^/PI^−^)	% Cholesterol Flow (M540^−^/PI^−^)	% PS Translocation (ANEX^−^/PI^−^)	% Caspase Activation (FLICA^−^/PI^−^)
Conventional	46.8 ± 14.6	29.6 ± 12.9	21.6 ± 11.7 ^a^	0.77 ± 0.50 ^a^	2.0 ± 1.65
Automated	57.2 ± 5.8	38.3 ± 11.9	35.0 ± 10.5 ^b^	0.53 ± 0.48 ^b^	1.21 ± 0.71

The results are shown as the mean ± SD (*n* = 12). The superscripts ^a,b^ indicate statistically significant differences (*p* < 0.05) between the two freezing techniques.

## Data Availability

The data supporting the findings of this study are available from the corresponding author upon reasonable request.

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
