# Peer review of "Slow Freezing of Preserved Boar Sperm: Comparison of Conventional and Automated Techniques on Post-Thaw Functional Quality by a New Combination of Sperm Function Tests"

_animals, 2023, doi:10.3390/ani13182826_

Round 1

Reviewer 1 Report

Article: Slow freezing for preserved boar sperm: comparison of conven-2 tional and automated techniques on the post-thaw functional 3 quality by a new combination of sperm function tests

Reference: animals-2472186

This article deals with the effect of the freezing curve on the seminal quality. Specifically, it compare the use of the styrofoam box (freezing in liquid nitrogen vapors) with the biofreezer which controls the freezing rate.

The experimental design is consistent with the stated objective. The battery of seminal analyzes is extensive. It remains to detail, in M&M, why some parameters such as: peroxynitrite levels, Superoxide anion levels, Phosphatidylserine translocation… The results are correctly exposed. Since such a complete analysis of the semen is made, the discussion does not explain parameters such as the Mitochondrial membrane potential. As indicated in the manuscript, this is not the first time that a similar study has been carried out in other species, and even in pigs. So the only novelty is to carry out a greater number of sperm tests. Therefore, why the biofreezer is better should be discussed more. With this I am not saying to write a longer discussion, but to delve deeper into the processes that can occur.

Other comment:

L103. Indicate de species. Number of animals

L195. Indicate Caspasa 3/7 assay

Table 1. Indicate the units

Figure 1. indicate the meaning of **

L343: remove the “6.” From the end of the sentence

L333-336. the conclusions have to be more concrete. delete this sentences

Author Response

Reviewer 1: This article deals with the effect of the freezing curve on the seminal quality. Specifically, it compares the use of the styrofoam box (freezing in liquid nitrogen vapors) with the biofreezer which controls the freezing rate.

Q 1. The experimental design is consistent with the stated objective. The battery of seminal analyzes is extensive. It remains to detail, in M&M, why some parameters such as: peroxynitrite levels, Superoxide anion levels, Phosphatidylserine translocation… The results are correctly exposed. Since such a complete analysis of the semen is made, the discussion does not explain parameters such as the Mitochondrial membrane potential. As indicated in the manuscript, this is not the first time that a similar study has been carried out in other species, and even in pigs. So the only novelty is to carry out a greater number of sperm tests. Therefore, why the biofreezer is better should be discussed more. With this I am not saying to write a longer discussion, but to delve deeper into the processes that can occur.

A 1. First, thank you very much for your comments. There is a precedent where both freezing techniques are compared (Baishya et al., 2018) however, the analyzes were carried out with bright field and fluorescence microscopy. We believe that our work provides novel information regarding sperm function, particularly those related to with cellular aging processes, therefore, include parameters that determine this process such as peroxynitrite and superoxide anion levels and their relationship with membrane components such as PS. All this using objective technologies such as the SCA system and flow cytometry. In relation to the biofreezer see the lines 266-272; 299-305 and 366-369.

Q 2. L103. Indicate de species. Number of animals

A 2. We have addressed this point; please see section 2.4

Q 3. L195. Indicate Caspasa 3/7 assay

A 3. We have addressed this point; please see section 2.7.6

Q 4. Table 1. Indicate the units

A 4. We have addressed this point; please see table 1

Q 5. Figure 1. indicate the meaning of **

A 5. We have addressed this point; please see figure 1 captions.

Q 6. L343: remove the “6.” From the end of the sentence

A 6. We have addressed this point; please see conclusion section.

Q 7. L333-336. the conclusions have to be more concrete. delete this sentences

A 7. We have addressed this point; please see conclusion section

Reviewer 2 Report

The manuscript “Slow freezing for preserved boar sperm: comparison of conventional and automated techniques on the post-thaw functional quality by a new combination of sperm function tests” aimed to compare both freezing techniques through a panel of evaluations of sperm function by CASA and flow cytometry technologies. The authors concluded that the conventional technique could be a cost-efficient alternative for boar sperm freezing. The study was well written, and the data well presented. I suggest a few small suggestions:

Abstract and introduction:

1. In abstract: The authors described the conventional technique but did not describe the automated technique. I suggest authors insert this information in the abstract.

2. The summary and abstract conclusions seemed inconsistent to me. I suggest the authors clarify this inconsistency and present both for the same purpose.

3. The single-paragraph introduction seems confusing to me. I suggest authors present the introduction divided into a few paragraphs.

4. In the introduction, insert the efficiency rates of the three techniques (vitrification, conventional freezing, and automated freezing) on sperm motility in pigs.

Material and methods:

1. Detail: is one ejaculate equivalent to one male and one repetition? How many ejaculates were used per subject?

2. What does n=20 d and =12 mean?

Results, discussion, and conclusion:

1. In “However, this effect was not observed in caspase 3 and 7 activation”. What?

2. In the discussion, better group the distribution of paragraphs.

3. In “One of the main limitations to implementing the use of an automated system is the high cost and maintenance of the freezing equipment”, so why did the authors rate it? It seems confusing to me and the end of the discussion.

Author Response

Reviewer 2: The manuscript “Slow freezing for preserved boar sperm: comparison of conventional and automated techniques on the post-thaw functional quality by a new combination of sperm function tests” aimed to compare both freezing techniques through a panel of evaluations of sperm function by CASA and flow cytometry technologies. The authors concluded that the conventional technique could be a cost-efficient alternative for boar sperm freezing. The study was well written, and the data well presented. I suggest a few small suggestions:

Abstract and introduction:

Q 1. In abstract: The authors described the conventional technique but did not describe the automated technique. I suggest authors insert this information in the abstract.

A 1. We have addressed this point; please see the abstract section. We include the rate of temperature drop during the critical period of freezing, which is when the freezing of the different cellular components actually occurs.

Q 2. The summary and abstract conclusions seemed inconsistent to me. I suggest the authors clarify this inconsistency and present both for the same purpose.

A 2. We have addressed this point; please see the abstract section. We only leave as a conclusion the fact that by controlling the descent curve certain sperm parameters are improved.

Q 3. The single-paragraph introduction seems confusing to me. I suggest authors present the introduction divided into a few paragraphs.

A 3. We have addressed this point; please see the introduction section.

Q 4. In the introduction, insert the efficiency rates of the three techniques (vitrification, conventional freezing, and automated freezing) on sperm motility in pigs.

A 4. We have addressed this point; please see the introduction section.

Material and methods:

Q 1. Detail: is one ejaculate equivalent to one male and one repetition? How many ejaculates were used per subject?

A 1. We have addressed this point; please see the section 2.4. For this study, twenty ejaculates from 8 different males were frozen/thawed.

Q 2. What does n=20 d and =12 mean?

A 2. Of those 20 ejaculates described in section 2,4, 12 were evaluated by flow cytometry and all the ejaculates were evaluated for motility.

Results, discussion, and conclusion:

Q 1. In “However, this effect was not observed in caspase 3 and 7 activation”. What?

A 1. We have addressed this point; please see the section 3.2.

Q 2. In the discussion, better group the distribution of paragraphs.

A 2. We have addressed this point; please see the section 4.

Q 3. In “One of the main limitations to implementing the use of an automated system is the high cost and maintenance of the freezing equipment”, so why did the authors rate it? It seems confusing to me and the end of the discussion.

A 3. We have addressed this point; please see at the of the section 4.